# Rational Design of Nitrogen-Doped Carbon Dots for Inhibiting *β*-Amyloid Aggregation

**DOI:** 10.3390/molecules28031451

**Published:** 2023-02-02

**Authors:** Hong Liu, Huazhang Guo, Yibin Fang, Liang Wang, Peng Li

**Affiliations:** 1Department of Neurovascular, Shanghai Fourth People’s Hospital, Tongji University School of Medicine, Shanghai 200434, China; liuhong0124@outlook.com; 2Department of Neurology, Shanghai East Hospital, Tongji University School of Medicine, Shanghai 200120, China; 3Institute of Nanochemistry and Nanobiology, School of Environmental and Chemical Engineering, Shanghai University, Shanghai 200444, China; guohuazhang@shu.edu.cn

**Keywords:** Alzheimer’s disease, carbon dots, nitrogen-doped, *β*-amyloid, aggregation

## Abstract

The fibrillization and abnormal aggregation of *β*-amyloid (A*β*) peptides are commonly recognized risk factors for Alzheimer’s disease (AD) brain, and require an effective strategy to inhibit the A*β* deposition and treat AD. Herein, we designed and synthesized nitrogen-doped carbon dots (N-CDs) as an A*β*-targeted probe, which exhibits the capacity of inhibiting the 1–42 A*β* (A*β*_1–42_) self-assembly in vitro. The N-CDs exhibited orange emission with an emission wavelength of 570 nm, which demonstrates their excellent optical properties with excitation-independent behavior. Meanwhile, the N-CDs have spherical morphologies with an average size of 2.2 nm, whose surface enriches the amino, carboxyl, and hydroxyl groups. These preparties are conducive to improving their biological water solubility and provide a large number of chemical bonds for further interaction with proteins. Contrary to this, the kinetic process, size evolutions, and morphologies changes of A*β*_1–42_ were inhibited in the presence of N-CDs in the determination of a thioflavin T assay, dynamic light scattering, transmission electron microscope, etc. Finally, the safety application of N-CDs on A*β*_1–42_-induced cytotoxicity was further demonstrated via in vitro cytotoxicity experiments. This work demonstrates the effective outcome of suppressing A*β* aggregation, which provides a new view into the high-efficiency and low-cytotoxicity strategy in AD theranostics.

## 1. Introduction

Alzheimer’s disease (AD), a common neurodegenerative disorder, is one of the major causative factors to induce progressive dementia [1]. Notably, AD is reported to affect more than 50 million people, with its prevalence continuing to grow in part because of the aging worldwide population [2]. The symptoms of the disease begin with mildly impaired memory function and evolve towards severe cognitive loss, inevitably terminating in complete incapacity and death [3]. Typically, its pathogenesis is characterized by extracellular aggregates of amyloid *β* (A*β*) plaques and intracellular neurofibrillary tangles of hyperphosphorylated Tau [4]. As a principal variant of the A*β* peptides in humans [5], A*β*_1–42_ contains all amino acid sequences and relatively accurately simulates the action of A*β* [6], but also serves as the most common variant in human cerebrospinal fluid [7]. Growing evidence suggests that there are currently no available treatments that can change the course of an illness or the rate of decline [8]. Clinically effective treatment methods include cholinesterase inhibitors for patients with any stage of AD and memantine for people with moderate-to-severe AD [9]. Nevertheless, these medications can only enhance the quality of life when prescribed at the appropriate time during an illness [10]. Thus, there is an urgent need to explore a novel pathway for preventing or treating AD.

Increasing attention has been paid to nanomedicine due to nanomaterials’ great biocompatibility [11,12,13], stable physiochemical properties [14,15,16], photoluminescence properties [17,18,19], edge effect [20], and low cytotoxicity [21,22]. Recent studies have exhibited promising results with regard to the probability of using carbon nanomaterials for amyloid fibrillogenesis [23,24]. The conjugated structure of graphene oxide helps it bind tightly to A*β* through hydrophobic interactions and π-π packing interactions. For instance, thioflavin-S-modified graphene oxide under infrared laser irradiation could dissociate amyloid aggregation due to its high near-infrared absorbance, indicating the possibility of the photothermal treatment of AD [25]. Beyond the graphene oxide, a nano-chaperone based on a mixed-shell polymeric micelle was applicable in selectively capturing A*β* peptides, thus inhibiting A*β* aggregation [26]. However, it cannot be neglected that the hampering effect of the blood-brain barrier (BBB) acts as an obstacle to the transport of these nanomaterials from the vascular compartment to the brain [27]. In addition, as amyloid plaques accompanied by high levels of metal ions (e.g., copper, zinc, iron), coordination compounds [28], gold nanoparticles [29], and metallosupramolecular complexes [30] were synthesized and positive effects were noted on their association of metal ions with *β*-amyloidosis. Nevertheless, the inhibitors containing metallic elements are toxic to the body at a high dose [31]. Fortunately, carbon dots (CDs) can be an ideal candidate, with greater biocompatibility and a nontoxic nature due to their ability to cross the BBB [32], in addition to the lack of metals [33]. Therefore, amongst all nanoparticle species, the employment of CDs can be inspiring news for inhibiting the aggregation of the A*β* peptide and envisioning its clinical use as an anti-AD drug.

Here, the newly discovered functions of nitrogen-doped carbon dots (N-CDs) inhibited A*β* aggregation was designed and prepared with 1,2,3-benzenetricarboxylic acid and o-phenylenediamine by a one-pot solvothermal method. The N-CDs display excellent optical performance and narrow size distribution. Furthermore, the kinetic process, size evolutions, and morphologies of A*β*_1–42_ with or without the presence of N-CDs were displayed by a thioflavin T (ThT) assay, dynamic light scattering (DLS), and transmission electron microscope (TEM), respectively. The morphology and spectroscopic characterizations of N-CDs were also shown. Moreover, the safety application of N-CDs on Aβ_1–42_-induced cytotoxicity was further demonstrated via in vitro cytotoxicity experiments. 

## 2. Results and Discussion

### 2.1. Characterization of N-CDs

To synthesize CDs, 1,2,3-benzenetricarboxylic acid and o-phenylenediamine were treated as precursors under a solvothermal condition at 180 °C for 12 h. The product solution was removed in large size and solvent, then transferred to the normal saline phase, and the pH was adjusted to 5 (Figure 1). The N-CDs presents an orange emission. As shown in Figure 2a, the UV-vis absorption and photoluminescence (PL) spectroscopy were explored. The N-CDs displayed an absorption peak at 420 nm and their absorption edge extend to 510 nm. The maximum photoluminescence excitation (PLE) and emission wavelength were 380 and 570 nm, respectively. The N-CDs demonstrated stable emission-independent PL behaviors from 360 to 540 nm excitation (Figure 2b). Furthermore, the N-CDs illustrated a monoexponential fluorescence lifetime of 3.21 ns (Figure 2c).

The morphology and structure were further represented by N-CDs. As illustrated in Figure 3a, the TEM image showed that the N-CDs has spherical morphologies with an average size of 2.2 nm. The D band (1362 cm^−1^) and G band (1580 cm^−1^) were found in its Raman spectrum (Appendix A), and the intensity ratio *I_D_*/*I_G_* was 0.73. The ratio indicates that N-CDs consist of a predominantly graphene structure and other disorder structures. [17,34] In addition, the surface properties of the N-CDs were further characterized by FT-IR and XPS spectra. The FT-IR spectrum (Figure 3b) of the N-CDs exhibited several peaks for O-H/N-H/C-H stretches at 2900–3410 cm^−1^, C=O, C=N, C-N, and C-O stretches at 1710, 1550, 1270, and 1110 cm^−1^, respectively. The summary of the FT-IR spectrum indicates the presence of N-C and O-C bonds from the amino, carboxyl, hydroxyl, and amide bond groups of N-CDs, which is further verified by its XPS measurement. The XPS survey show the N-CDs’ composition of C (77.06 at.%), N (10.31 at.%), and O (12.63 at.%) in Figure 3c. The XPS C1s spectrum (Figure 3d) displayed sp^2^ C (C-C/C=C) at 284.8 eV, sp^3^ C (including C-N and C-O) at 285.58 and 286.76 eV, a carbonyl group C (C=O or C=N) at 289.31 eV, and π-π* satellite at 291.98 eV. As shown in Figure 3e, the XPS N1s spectrum revealed the four N sources, including pyridinic N (398.56 eV), amine N (399.01 eV), pyrrolic N (400.11 eV), and graphitic N (400.81 eV). In Figure 3f, the XPS O1s spectrum was distributed to C=O, O=C-NH, and C-O, at 530.88, 532.30, and 533.82 eV, respectively. The dominant hydroxyl, amino, and carboxyl groups on the N-CDs’ surface support better aqueous solubility and further interact with protein via chemical bonds. Previous reports had demonstrated that inhibition can occur by distinct binding patterns between Aβ monomers and the surfaces of the CDs with hydrophilic/hydrophobic groups (Appendix A) [35,36].

### 2.2. Inhibition on Aβ_1–42_ Aggregation Self-Assembly with N-CDs

Nanomaterial-based approaches could offer promising directions in addressing the challenges in current therapeutic/diagnostic bio-reagent applications [37,38]. In the present study, the modulation effects of the special N-CDs on the amyloid peptide assemblies are presented according to the ThT assay, DLS, and TEM. As shown in Figure 4a, the effect of N-CDs on the assembly of A*β*_1–42_ peptides was first employed by a ThT fluorescence assay, which is a widely used method for monitoring amyloid aggregation and the fluorescence intensity increases with the augment of conjugates in the normal condition [39]. Compared to A*β*_1–42_ alone (the first column), the fluorescence intensity was significantly lower when A*β*_1–42_ peptides were incubated with N-CDs after 24 h incubation (second to fifth column). Furthermore, the value was even lower after increasing the dosage of N-CDs.

The side effect of CDs on modulating the A*β*_1–42_ peptide assembly was explored through DLS measurement. As shown in Figure 4b, the solution of A*β*_1–42_ was incubated with and without N-CDs in a PBS buffer at 37 °C for 24 h. The pure A*β*_1–42_ has an obvious agglomeration phenomenon after 24 h. However, the aggregation of the A*β*_1–42_ peptide was inhibited in the presence of N-CDs after 24 h incubation compared with the free A*β*_1–42_ peptide (67.58 ± 15.84 vs. 364.20 ± 138.90, *p* < 0.05). Our results clearly show at least a fivefold reduction owing to N-CDs. Such an obvious phenomenon proves that N-CDs acting as an Aβ-targeted probe is feasible.

To further verify the inhibitory effect of N-CDs, the morphology of the A*β*_1–42_ peptide in the presence or absence of N-CDs was also investigated in Figure 5. Figure 5a shows the initial morphology of the A*β*_1–42_ peptide. In the absence of N-CDs, A*β*_1–42_ peptides formed a typical structure for amyloid fibrils (Figure 5b), whereas the formation of the fibril was remarkably lower in the presence of N-CDs compared with the absence group (Figure 5c). The TEM data further supported the results with the ThT fluorescence assay and DLS, indicating that N-CDs could conspicuously influence the aggregation of the A*β*_1–42_ peptide. Previous studies also showed that various carbon materials including fullerene, carbon nanotubes, and graphene oxide were applied to inhibit the A*β*_1–42_ peptide aggregation [40,41]. As a newly prepared nano-material, we proved that the inhibiting affectivity of the special N-CDs plays a dominant role in the process of inhibiting the aggregation based on the above analysis in this work.

### 2.3. Aβ-Induced Cell Viability Was Reversed by N-CDs

To explore whether N-CDs can improve cell viability induced by A*β*_1–42_, Figure 6 demonstrates that the cellular reduction in the A*β*_1–42_ group was reversed by N-CDs as measured by MTT assay in a dose-dependent manner. As compared to the control, the cell viability was lower in the purely A*β*_1–42_-treated group (64.81% ± 3.41%, *p* < 0.001, third column), indicating that A*β*_1–42_ causes a noticeable negative effect on N2a cells. Contrary to this, while N-CDs were added into the plates pre-incubated with A*β*_1–42_ for 24 h, the cell viability was significantly higher (fourth to seventh columns). Notably, the ability of N-CDs to increase cell viability (95.82% ± 0.79% vs. 64.81% ± 3.41%, *p* < 0.001) suggested that the N-CDs can afford to be an available drug target in blocking A*β*_1–42_ assembly.

## 3. Materials and Methods

### 3.1. Materials

1,2,3-benzenetricarboxylic acid hydrate was purchased from TCI. o-phenylenediamine, ethanol, and hydrochloric acid (HCl) were acquired from Adams (Shanghai Titan Scientific Co., Ltd., Shanghai, China). These chemical reagents were utilized without further purification. A*β*_1–42_ peptide was purchased from Sigma (Sigma-Aldrich, St. Louis, MO, USA) and re-suspend according to the manufacturer’s protocols.

### 3.2. Synthesis of N-CDs

Briefly, a 1:1 mass ratio of 1,2,3-benzenetricarboxylic acid (50 mg, 0.24 mmol) and o-phenylenediamine (50 mg, 0.46 mmol) were dispersed in 10 mL ethanol by ultrasound. The mixture solution was then transferred to 25 mL of a poly-(tetrafluoroethylene) (Teflon)-lined autoclave and reacted at 180 °C for 12 h. The primary product was filtered and the solvent was removed. The normal saline was then added to the above, and adjusted to pH = 5 with 0.1 M HCl. The final concentration of the N-CDs solution is 6 mg/mL. Meanwhile, the N-CDs powder was acquired for further characterization after purifying by dialysis (MWCO 3500) for one week.

### 3.3. Thioflavin T (ThT) Assay

ThT powder (Shanghai Aladdin Biology, Shanghai, China) was dissolved into phosphate buffer saline (PBS) at a concentration of 1 mM and filtered by a 0.22 mm filter membrane before use. The re-suspended A*β*_1–42_ peptide solution was then re-dissolved into PBS buffer at a concentration of 100 µM. The A*β*_1–42_ solutions were then incubated with ThT solution and N-CDs at a concentration of 6, 3, 1.5, and 0.75 mg/mL 37 °C for 24 h. A*β*_1–42_ solutions prepared with the same amount of PBS were used as the control. The ThT fluorescence intensity was recorded at the beginning and end of incubation by a microplate reader (Spectramax M5, San Jose, CA, USA) at an excitation and emission wavelength of 450 and 482 nm, respectively.

### 3.4. Cell Viability Detected by MTT Assay

Mouse neuroblastoma Neuro-2a cells were first cultured in Dulbecco’s Modified Eagle’s Medium (DMEM) added with 10% fetal bovine serum (FBS) (Shanghai Fuheng Biology, Shanghai, China) at 37 °C in a humidified (5% CO_2_, 95% air) incubator. Cells were seeded into 96-well microplates with a density of 8000 cells per well, cultured overnight, and then treated with A*β*_1–42_ (100 µM), N-CDs (6 mg/mL), or A*β*_1–42_ (100 µM)/N-CDs mixtures at a concentration of 6, 3, 1.5, and 0.75 mg/mL. After being incubated for another 24 h, a 3-(4,5-dimethylthiazol-2-yl)-2,5-diphenyltetrazolium bromide (MTT) assay (Beyotime Institute of Biotechnology, Shanghai, China) was added to assess the cell viability according to the manufacturer’s protocols. 

### 3.5. Characterization

PL and UV-vis absorption spectra were characterized by Horiba Duetta. The time-resolved PL spectra were obtained with a Horiba FluoroMax with a 455 nm laser. A transmission electron microscopy (TEM) image was performed by using a Japan Hitachi HT7700. The Raman spectrum was achieved by an Anton-Paar Cora 5001 with 785 nm of laser wavelength. The Fourier transform infrared (FT-IR) spectrum was characterized with a Thermo Scientific iS50 FT-IR. The X-ray photoelectron spectroscopy (XPS) spectrum was acquired using a Thermo ESCALAB 250Xi spectrometer. Dynamic light scattering (DLS) is a powerful tool used to monitor particle size evolutions. A*β*_1–42_ peptides were mixed with 6 mg/mL N-CDs incubated for 24 h at 37 °C and then the size distribution was measured by a ZetasizerNano ZS nanoparticle size analyzer (Malvern Instruments Ltd., Malvern, UK). A*β*_1–42_ solutions incubated with PBS were also utilized as controls. After measuring the effect of N-CDs on A*β*_1–42_ fibrosis by ThT and DLS as shown previously, the morphologies of the A*β*_1–42_ aggregates incubated with or without N-CDs were observed and visualized by TEM (Japan Hitachi HT7700, Japan, Tokyo).

### 3.6. Statistical Analysis

All analyses were performed using the SPSS software (version 26.0; SPSS, Chicago, IL, USA). Data are expressed as the mean ± standard error of the mean of at least three biological replicates for each experiment. Statistical differences were analyzed by Student’s *t*-test for comparisons between two groups or by ANOVA followed by Tukey’s multiple comparison post hoc test for comparisons among more than two groups. A *p*-value < 0.05 was considered statistically significant.

## 4. Conclusions

Prompted by the need to pursue an effective treatment of AD, many inhibitors against A*β* fibration and cytotoxicity were explored [42,43]. As a novel carbon nanomaterial, CDs have received significant attention in materials science [44,45] and biomedicine [46,47,48]. In summary, we designed the N-CDs prepared by 1,2,3-benzenetricarboxylic acid and o-phenylenediamine for inhibiting A*β* aggregation. The N-CDs exhibited orange emission with an emission wavelength of 570 nm, which demonstrates its excellent optical properties with excitation-independent behavior. Meanwhile, the N-CDs has a spherical morphology with an average size of 2.2 nm, whose surface enriches the amino, carboxyl, and hydroxyl groups. These properties are conducive to improving its biological water solubility and providing a large number of chemical bonds for further interaction with proteins. Contrary to this, the kinetic process, size evolutions, and morphology changes of A*β*_1–42_ were inhibited in the presence of N-CDs in the determination of thioflavin T assay, dynamic light scattering, transmission electron microscope, etc. Finally, the safety application of N-CDs on A*β*_1–42_ was further demonstrated via in vitro cell viability experiments. Those discoveries make it reasonable to speculate that the N-CDs can inhibit A*β*_1–42_ aggregation and alleviate A*β*-induced cytotoxicity. Our work improves the probability of the use of the particular nanostructures on A*β* monomers aggregating into fibrils in the future.

## Figures and Tables

**Figure 1 molecules-28-01451-f001:**
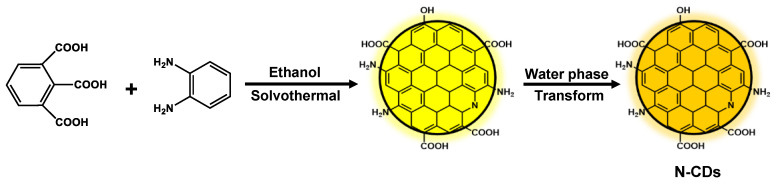
Schematic route for the synthesis of N-CDs.

**Figure 2 molecules-28-01451-f002:**
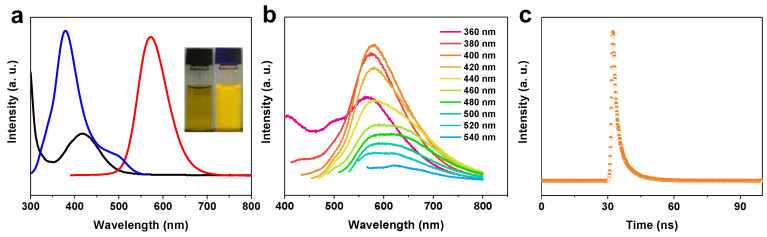
(**a**) The UV-vis, PL, and PLE spectra, (**b**) the PL spectra at different excitation wavelengths, and (**c**) the time-resolved PL spectrum of N-CDs.

**Figure 3 molecules-28-01451-f003:**
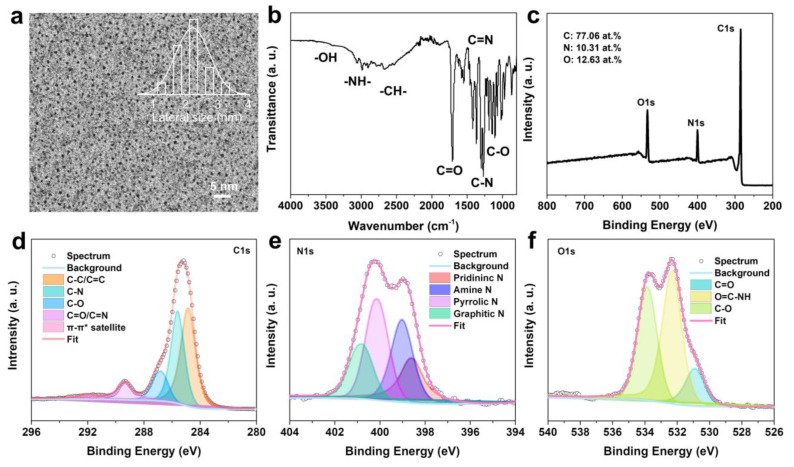
(**a**) TEM image and its corresponding lateral size distribution, (**b**) FT-IR spectrum, (**c**) survey XPS spectrum, (**d**) high-resolution C1s spectrum, (**e**) high-resolution N1s spectrum, and (**f**) high-resolution O1s spectrum of N-CDs.

**Figure 4 molecules-28-01451-f004:**
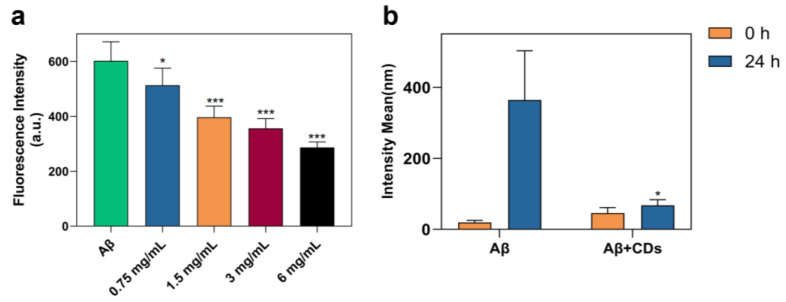
A*β*_1–42_ aggregation was inhibited by N-CDs. (**a**) The thioflavin T fluorescence intensity in the presence or absence of N-CDs at the concentration of 0.75, 1.5, 3, and 6 mg/mL after 24 h incubation. *** *p* < 0.001, * *p* < 0.05 vs. A*β*_1–42_ alone (1st column). (**b**) DLS spectra of A*β*_1–42_ peptide with or without N-CDs (6 mg/mL) at the beginning and after 24 h incubation. The aggregation of the A*β*_1–42_ peptide was inhibited in the presence of N-CDs after 24 h incubation compared with the free A*β*_1–42_ peptide (* *p* < 0.05).

**Figure 5 molecules-28-01451-f005:**

TEM images of (**a**) A*β*_1–42_ peptide at 0 h, (**b**) A*β*_1–42_ peptide incubated for 24 h, and (**c**) A*β*_1–42_ peptide incubated with 6 mg/mL N-CDs for 24 h.

**Figure 6 molecules-28-01451-f006:**
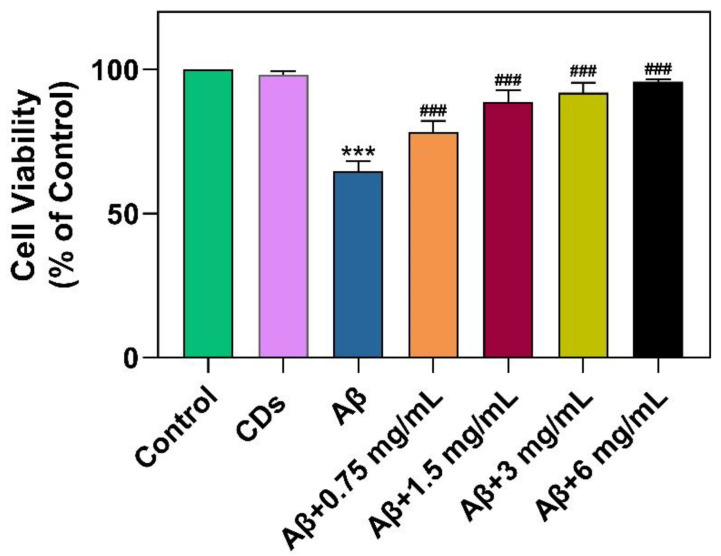
Effect of A*β*_1–42_ peptide (100 µM), N-CDs (6 mg/mL), or A*β*_1–42_ (100 µM) mixed with N-CDs at the concentration of 0.75, 1.5, 3, and 6 mg/mL on the viability of N2a cells after 24 h of incubation. (*** *p* < 0.001 vs. Control; ^###^
*p* < 0.001 vs. purely A*β*_1–42_-treated group).

## Data Availability

All data generated or analyzed during this study are included in this published article and its Appendix A.

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
