# Peer review of "Rational Design of Nitrogen-Doped Carbon Dots for Inhibiting β-Amyloid Aggregation"

_molecules, 2023, doi:10.3390/molecules28031451_

Round 1
Reviewer 1 Report
Carbon dots (CDs) are widely used in the biomedical field due to their excellent water solubility, low toxicity, satisfying biocompatibility, and good tissue permeability. The authors designed and synthesized nitrogen-doped carbon dots (N-CDs), which can inhibit Aβ aggregation. The work is very meaningful.
(1) The line 97 “And the D band (1362 cm−1) and G band (1580 cm−1) were found in 97its Raman spectrum (Figure S1), and the intensity ratio ID/IG of 0.73.” The ratio 0.73 means what?
(2) Compared to other carbon nanomaterials, such as graphene oxide (whose surface enriches epoxy, hydroxyl, and carboxyl group), inhibitory effect of the nitrogen-doped carbon dots (enriches amino, carboxyl, hydroxyl, and amide bond groups) on Aβ aggregation is better? Whether amino and amide bond groups play an important role? Need to discuss this.
(3) The line 111 “The dominant hydroxyl, amino, and carboxyl groups on the N-CDs’ surface support better aqueous solubility and further interact with protein via chemical bonds.” The aqueous solubility of N-CDs is better than what? To our knowledge, carbon dots mainly interact with proteins via hydrogen-bond, hydrophobic, and π-π packing interactions rather than chemical bonds.
(4) The Aβ aggregation process consists of lag, elongation and steady phases. Which stage do N-CDs mainly affect?

Reviewer 2 Report
This is an interesting study that focuses on the prevention of beta-amyloid aggregation. However, it needs some work to be published. It is necessary to introduce statistical processing of the results. There should be a corresponding section in Methods. Figures 4 and 6 should have statistical validity criteria. The number of repetitions must be specified. In Figure 6, it is necessary to indicate the data on the effect of nanodots on cells in the absence of beta-amyloid. Everywhere in the sections about cells, the concentration of beta-amyloid must be indicated.
Round 2
Reviewer 1 Report
Thank you for clarifications and amendments.
Author Response
Thank you again for your kind suggestions.
Reviewer 2 Report
The caption on the y-axis and the legend for Figure 6 must be corrected. This figure shows cell viability. The y-axis must be in percent, i.e. 100%, not 1.0. In the legend it is necessary to write "Effect of Abet, ...,.... on the viability of H2a cells after 24 hours of incubation...". In the text in Section 2.3, it is necessary to replace cytotoxicity, cell death, etc. with cell viability everywhere.
